# CrossTransformers: spatially-aware few-shot transfer

**Carl Doersch**[*]  **Ankush Gupta**[*]  **Andrew Zisserman**[*†]

[*] DeepMind, London  [†] VGG, Department of Engineering Science, University of Oxford

## Abstract

Given new tasks with very little data—such as new classes in a classification problem or a domain shift in the input—performance of modern vision systems degrades remarkably quickly. In this work, we illustrate how the neural network representations which underpin modern vision systems are subject to *supervision collapse*, whereby they lose any information that is not necessary for performing the training task, including information that may be necessary for transfer to new tasks or domains. We then propose two methods to mitigate this problem. First, we employ self-supervised learning to encourage general-purpose features that transfer better. Second, we propose a novel Transformer based neural network architecture called CrossTransformers, which can take a small number of labeled images and an unlabeled query, find coarse spatial correspondence between the query and the labeled images, and then infer class membership by computing distances between spatially-corresponding features. The result is a classifier that is more robust to task and domain shift, which we demonstrate via state-of-the-art performance on Meta-Dataset, a recent dataset for evaluating transfer from ImageNet to many other vision datasets. Code available at: https://github.com/google-research/meta-dataset.

## 1  Introduction

General-purpose vision systems must be adaptable. Home robots must be able to operate in new, unseen homes; photo-organizing software must recognize unseen objects (e.g., to find examples of "my sixth-grade son's abstract art project"); industrial quality-assurance systems must spot defects in new products. Deep neural network representations can bring some visual knowledge from datasets like ImageNet [68] to bear on different tasks beyond ImageNet [15, 32, 62], but empirically, this requires a non-trivial amount of labeled data in the new task. With too little labeled data, or for a large change in distribution, such systems empirically perform poorly.

Research on meta-learning directly benchmarks adaptability. At training time, the algorithm receives a large amount of data and accompanying supervision (e.g., labels). At test time, however, the algorithm receives a series of *episodes*, each of which consists of a small number of datapoints from a *different distribution* than the training set (e.g., a different domain or different classes). Only a subset of this data has the accompanying supervision (called the *support set*); the algorithm must make predictions about the rest (the *query set*). Meta-Dataset [86] is particularly relevant for vision, as the challenge is few-shot fine-grained image classification. The training data is a subset of ImageNet classes. At test time, each episode either contains images from the other ImageNet classes, or from one of nine other visually distinct fine-grained recognition datasets. The algorithm must rapidly adapt its representations to the new classes and domains.

Simple centroid-based algorithms like Prototypical Nets [17, 76] are near state-of-the-art on Meta-Dataset, achieving around 50% accuracy on the held-out ImageNet classes in Meta-Dataset's validation set (chance is roughly 1 in 20). An equivalent classifier trained on those validation classes can achieve roughly 84% accuracy on the same challenge. What accounts for the enormous discrepancy

**Query**

**Nearest Neighbors**

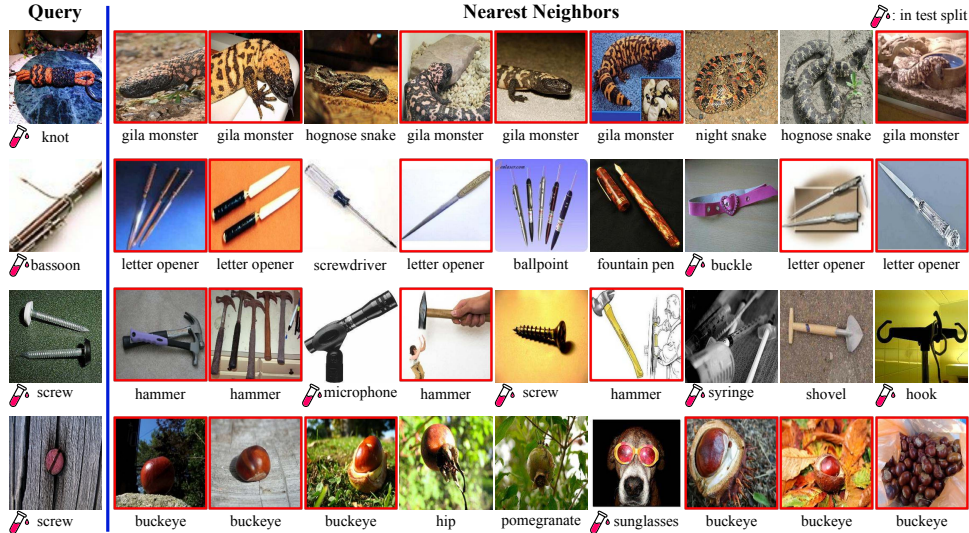

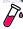

Figure 1: **Illustration of supervision collapse with nearest neighbors.** In each row, the leftmost image is a query taken from the Meta-Dataset ImageNet *test* classes, and the rest are the top 9 nearest neighbors from *both training and test* support set classes, using the embedding learned by a Prototypical Net (training details in the extended version [1]). Images belonging to the test split are indicated by a 🧪 near the bottom left corner; rest are from the training split. For a simple classifier to work well on these test classes, semantically similar images should have similar representations, and so we hope the nearest neighbors would come from the same—or semantically similar—classes. Instead, we observe that only 5% of matches for test-set queries are from the same class as the query. Furthermore, many matches are all from the *same incorrect training class* (highlighted in red). We see a knot is matched with several gila monsters (and other reptiles); a bassoon with letter openers (and pens); a screw with hammers; another screw with buckeyes. The errors within that wrong class often have widely different appearance: for example, the bottom-most screw is matched with single buckeyes and also a pile of buckeyes. One interpretation is that the network picks up on image patterns during training that allow images of each class to be tightly grouped in the feature space, minimizing other ways that the image might be similar to other classes in preparation for a confident classification. For out-of-domain samples, the network can then overemphasize a spurious image pattern that suggests membership in one training-set class. This is the consequence of supervision collapse, where image patterns that might help make the correct associations are lost.

between performance on within-distribution samples and out-of-distribution samples? We hypothesize that because the neural network backbone of Prototypical Nets is designed for classification, they do just this: represent *only* an image's (training-set) class, and discard information that might help with out-of-distribution classes. Doing so minimizes the losses for many meta-learning algorithms, including Prototypical Nets. We call this problem *supervision collapse*, and illustrate it in Figure 1.

Our first contribution is to explore using self-supervision to overcome supervision collapse. We employ SimCLR [16], which learns embeddings that discriminate between every image in the dataset while maintaining invariance to transformations (e.g., cropping and color shifts), thus capturing more than just classes. However, rather than treat SimCLR as an auxiliary loss, we reformulate SimCLR as "episodes" that can be classified in the same manner as a training episode.

Our second contribution is a novel architecture called *CrossTransformers*, which extends Transformers [88] to few-shot fine-grained classification. Our key insight is that objects and scenes are generally composed of smaller parts, with *local* appearance that may be similar to what has been seen at training time. The classical example of this is the *centaur* that appeared in several early papers on visual representation [12, 41, 89], where the parts from the human and horse composed the centaur.

CrossTransformers operationalize this insight of (i) local part-based comparisons, and (ii) accounting for spatial alignment, resulting in a procedure for comparing images which is more agnostic to the underlying classes. In more detail, first a coarse alignment between geometric or functional parts in the query- and support-set images is established using attention as in Transformers. Then, given this alignment, distances between corresponding local features are computed to inform classification. We demonstrate this improves generalization to unseen classes and domains.

In summary, our contributions in this paper are: (i) We improve the robustness of our local features with a self-supervised technique, modifying the state-of-the-art SimCLR [16] algorithm. (ii) We propose the CrossTransformer, a network architecture that is spatially aware and performs few-shot classification using more local features, which improves transfer. Finally, (iii) we evaluate and ablate how the choices in these algorithms impact Meta-Dataset [86] performance, and demonstrate state-of-the-art results on nearly every dataset within it, often by large margins.

## 2 Related Work

**Few-shot image classification.** Few-shot learning [26, 36, 47, 54] has recently been primarily addressed in the meta-learning framework [58, 72, 81], where a model learns an update rule for the parameters of a base-learner model [6, 7, 73] through a sequence of training episodes [80, 90]. The meta-learner either learns to produce new parameters directly from the new data [9, 34, 57, 63, 65, 73, 74], or learns to produce an update rule to iteratively optimize the base learner to fit the new data [2, 6, 8, 39, 64, 99]. [28, 52, 60] do not use any explicit meta-learner model, but instead unroll the base-learner gradient updates and optimize for model initializations which generalize well on novel tasks. Matching-based methods [29, 76, 79, 91] instead learn representations for similarity functions [11, 17, 18, 45, 83], in the hope that the similarities will generalize to new data. CrossTransformers fall in this category, and share much of their architecture with Prototypical Nets [76].

**Attention for few-shot learning.** CrossTransformers attend [4] individually over each class's support set to establish local correspondences, whereas Matching Networks [91] attend over the whole support set to "point to" matching instances. [55] extend this idea to larger contexts using temporally dilated convolutions [87]. In the limit, attention over long-term experiences accumulated in memories [44, 57, 70, 77] can augment more traditional learning.

**Correspondences for visual recognition.** CrossTransformers perform classification by matching more local parts. Discriminative parts [5, 10, 23, 35, 43, 84] and visual words [75, 102] have a rich history, and have found applications in deformable-parts models [27], classification [71, 102], and retrieval [13, 85]. Part-based correspondences for recognition [105] have been particularly successful in fine-grained face retrieval and recognition [49, 97]. CrossTransformers establish soft correspondences between pixels in the query- and support-set images; such dense pairwise interactions [94] have recently proved useful for generative networks [101], semantic matching [67] and tracking [92]. [50] learns spatially dense classifiers for few-shot classification, but pools the spatial dimensions of the prototypes, and hence does not have a notion of correspondence.

**Self-supervised learning for few-shot.** Our work on SimCLR episodes inherits from a line of self-supervised learning research, which typically deal with transfer from pretext tasks to semantic ones and must therefore represent more than their training data [3, 14, 16, 22, 25, 31, 37, 48, 61, 103, 104]. Some recent works [30, 78] demonstrate that this can improve few-shot learning, although these use self-supervised auxiliary losses rather than integrating self-supervised instance discrimination [16, 25, 56, 82, 96] into episodic training. Also particularly relevant are methods that use self-supervision for correspondence [42, 51, 92], which may in future work improve the correspondences that CrossTransformers use.

## 3 Stopping Collapse: SimCLR Episodes and CrossTransformers

We take a two-pronged approach to dealing with the supervision collapse problem. Modern approaches to few-shot learning typically involve learning an embedding for each image, followed by a classifier that aggregates information across an episode's support set in order to classify the episode's queries. Our first step aims to use self-supervised learning to improve the embedding so it expresses information beyond the classes, in a way that is as algorithm-agnostic as possible. Once we have these embeddings, we build a classifier called a CrossTransformer. CrossTransformers use Prototypical Nets [76] as a blueprint, chosen due to their simplicity and strong performance; the main modification is to aggregate information in a spatially-aware way. We begin by reviewing Prototypical Nets, and then describe the two approaches.

Prototypical Nets are *episodic learners*, which means training is performed on the same kind of episodes that will be presented at test time: a query set $Q$ of images, and a support set $S$ which can be partitioned into classes $c \in \{1, 2, \ldots, C\}$: each $S^c = \{x_i^c\}_{i=1}^N$ is composed of $N$ example images $x_i^c$. Prototypical Nets learn a distance function between the query and each subset $S^c$. Both the query- and support-set images are first encoded into a $D$-dimensional representation $\Phi(x)$, using a shared ConvNet $\Phi : \mathbb{R}^{H \times W \times 3} \mapsto \mathbb{R}^D$, where $H, W$ are the height and width respectively. Then a "prototype" $\boldsymbol{t}^c \in \mathbb{R}^D$ for the class $c$ is obtained by averaging the representations of the support set $S^c$, $\boldsymbol{t}^c = \frac{1}{|S^c|} \sum_{x \in S^c} \Phi(x)$. Finally, a distribution of classes is obtained using softmax over the distances between the query image and class prototypes: $p(y = c | x_q) = \frac{\exp(-d(\Phi(x_q), \boldsymbol{t}^c))}{\sum_{c'=1}^C \exp(-d(\Phi(x_q), \boldsymbol{t}^{c'}))}$. In practice, the distance function $d$ is fixed to be the squared Euclidean distance $d(x_q, S^c) = ||\Phi(x_q) - \boldsymbol{t}^c||_2^2$. The learning objective is to train the embedding network $\Phi$ to maximize the probability of the correct class for each query.

## 3.1 Self-supervised training with SimCLR

Our first challenge is to improve the neural network embedding $\Phi$: after all, if these features have collapsed to represent little information beyond the classes, then a subsequent classifier cannot can recover this information. But how can we train features to represent things beyond labels when our only supervision is the labels? Our solution is self-supervised learning, which invents "pretext tasks" that train representations *without* labels [22, 25], and better yet, has a reputation for representations that transfer beyond this pretext task. Specifically we use SimCLR [16], which uses "instance discrimination" as a pretext task. It works by applying random image transformations (e.g., cropping or color shifts) twice to the same image, thus generating two "views" of that image. Then it trains the network so that representations of the two views of the same image are more similar to each other than they are to those of different images. Empirically, networks trained in this way become sensitive to semantic information, but also learn to discriminate between different images *within a single class*, which is useful for combating supervision collapse.

While we could treat SimCLR as an auxiliary loss on the embedding, we instead reformulate SimCLR as episodic learning, so that the technique can be applied to all episodic learners with minimal hyper-parameters. To do this, we randomly convert 50% of the training episodes into what we call *SimCLR episodes*, by treating every image as its own class. For clarity, we will call the original episodes that have not been converted SimCLR episodes *MD-categorization episodes*, to emphasize that they use the original categories from Meta-Dataset. Specifically, let $\rho(\cdot)$ be SimCLR's (random) image transformation function, and let $S = \{x_i\}_{i=1}^{|S|}$ be a training support set. We generate a SimCLR episode by sampling a new support set, transforming each image in the original support set $S' = \{\rho(x_i)\}_{i=1}^{|S|}$, and then generating query images by sampling other transformations from the same support set: $Q' = \{\rho(\text{random\_sample}(S))\}_{i=1}^{|Q|}$, where $\text{random\_sample}$ just takes a random image from the set.[1] The original query set $Q$ is discarded. The label for an image in the SimCLR episode is its index in the original support set, resulting in an $|S|$-way classification for each query. Note that for a SimCLR episode, the 'prototypes' in Prototypical Nets average over just a single image, and therefore the Prototypical Net loss can be written as $\frac{\exp(-d(\Phi(\rho(x_q)), \Phi(\rho(x_q))))}{\sum_{i=1}^n \exp(-d(\Phi(\rho(x_q)), \Phi(\rho(x_i))))}$. If we define $d$ as the cosine distance rather than Euclidean, this loss is identical to the one used in SimCLR.

## 3.2 CrossTransformers

Given a query image $x_q$ and a support set $S^c = \{x_i^c\}_{i=1}^N$ for the class $c$, CrossTransformers aim to build a representation which enables local part-based comparisons between them.

CrossTransformers start by making the image representation a spatial tensor, and then assemble *query-aligned* class prototypes by putting the support-set images $S^c$ in correspondence with the query image. The distance between the query image and the query-aligned prototype for each class is then computed and used in a similar way to Prototypical Nets. In practice, we establish soft correspondences using attention [4] based Transformers [88]. In contrast, Prototypical Nets use flat vector representations which lose the location of image features, and have a fixed class prototype which is independent of the query image.

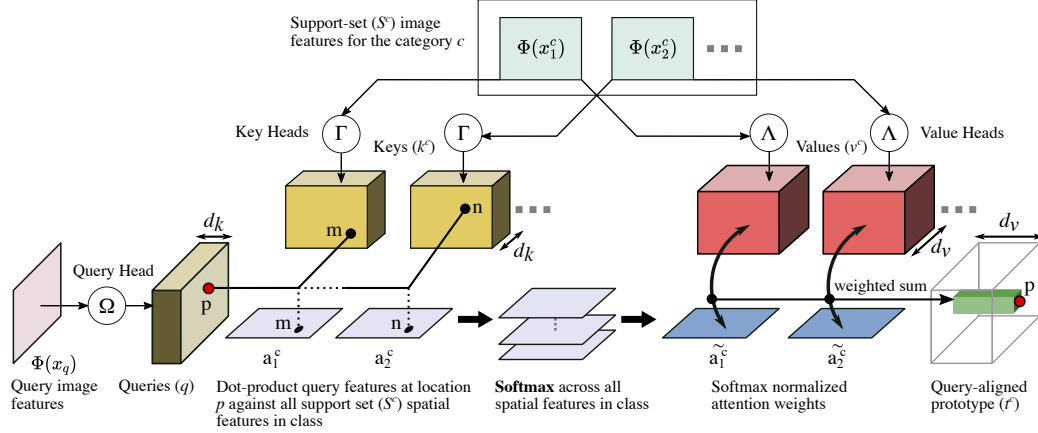

Figure 2: **CrossTransformers.** Construction of *query-aligned* class prototype vector $t_p^c$ for the class $c$ and the query image $x_q$, focusing on the spatial location $p$ in $x_q$. The query vector $q_p$ is compared against keys $k^c$ from all spatial locations in the support set $S^c$ to obtain attention scores $a^c$, which are softmax normalized before being used to aggregate the values $v^c$ for the aligned prototype vector $t_p^c$.

Concretely, CrossTransformers remove the final spatial pooling in Prototypical Nets' embedding network $\Phi(\cdot)$, such that the spatial dimensions $H', W'$ are preserved: $\Phi(x) \in \mathbb{R}^{H' \times W' \times D}$. Following Transformers, key-value pairs are then generated for each image in the support set using two independent linear maps: the key-head $\Gamma : \mathbb{R}^D \mapsto \mathbb{R}^{d_k}$, and the value-head $\Lambda : \mathbb{R}^D \mapsto \mathbb{R}^{d_v}$ respectively. Similarly, the query image features $\Phi(x_q)$ are embedded using the query-head $\Omega : \mathbb{R}^D \mapsto \mathbb{R}^{d_k}$. Dot-product attention scores are then obtained between keys and queries, followed by softmax normalization across all the images and locations in $S^c$. This attention serves as our coarse correspondence (see example attention visualizations in Figure 3 and the extended version [1]), and is used to aggregate the support-set features into alignment with the query. This process is visualized in Figure 2.

Mathematically, let $\boldsymbol{k}_{jm}^c = \Gamma \cdot \Phi(x_j^c)_m$ be the key for the $j^{\text{th}}$ image in the support set for class $c$ at spatial position $m$ (index over the two dimensions $H', W'$), and similarly let $\boldsymbol{q}_p = \Omega \cdot \Phi(x_q)_p$ be the query vector at spatial position $p$ in the query image $x_q$. The attention $\tilde{a}_{jmp}^c \in \mathbb{R}$ between the two is then obtained as:

$$\tilde{a}_{jmp}^c = \frac{\exp(a_{jmp}^c / \tau)}{\sum_{i,n} \exp(a_{inp}^c / \tau)}, \qquad \text{where} \quad a_{jmp}^c = \boldsymbol{k}_{jm}^c \cdot \boldsymbol{q}_p, \quad \text{and} \quad \tau = \sqrt{d_k}. \qquad (1)$$

Next, the aligned prototype vector $\boldsymbol{t}_p^c$ corresponding to spatial location $p$ in the query is obtained by aggregating the support-set values $\boldsymbol{v}_{jm}^c = \Lambda \cdot \Phi(x_j^c)_m$ using the attention weights above:

$$\boldsymbol{t}_p^c = \sum_{jm} \tilde{a}_{jmp}^c \boldsymbol{v}_{jm}^c \qquad (2)$$

Finally, squared Euclidean distances between aligned local features from the above prototype and corresponding query image values $\boldsymbol{w}_p = \Lambda \cdot \Phi(x^q)_p$ are aggregated as below. This scalar distance acts as a negative logit for a distribution over classes as in Prototypical Nets.

$$d(x_q, S^c) = \frac{1}{H'W'} \sum_p ||\boldsymbol{t}_p^c - \boldsymbol{w}_p||_2^2 \qquad (3)$$

Note we apply the same value-head $\Lambda$ to both the query and support-set images. This ensures that the CrossTransformer behaves somewhat like a distance. That is, imagine a trivial case where, for one class, all images in $S^c$ are identical to $x_q$. We would want $d(x_q, S^c)$ to approach 0 even if the network is untrained, or if these images are highly dissimilar from those used for training. Sharing $\Lambda$ between the support and query sets helps accomplish this: in fact, if $\tilde{a}_{jmp}^c$ is 1 where $p = m$ and 0 elsewhere for all $j$, then $d(x_q, S^c)$ will be identically 0 under this architecture, no matter the network

| Query | Correspondence in support set | Query | Correspondence in support set |
|---|---|---|---|

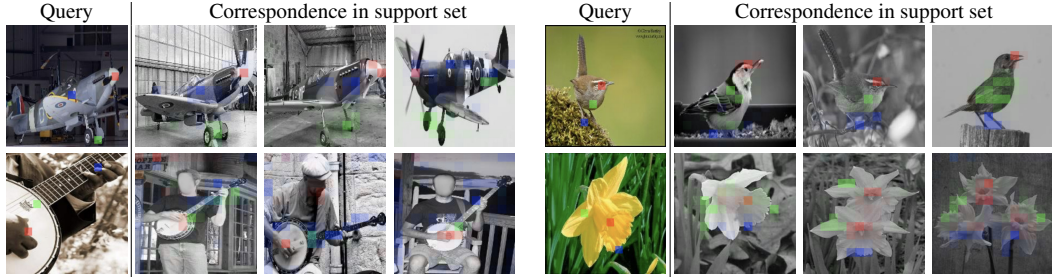

Figure 3: **Visualization of the attention** $\tilde{a}$. We show four query images, along with three support-set images for each. Within each query image, we choose three spatial locations (red, green, and blue squares), and plot the CrossTransformer attention weights for each one in the corresponding color (brighter colors mean higher weight). The four examples are from Aircraft, CU-Birds, VGG Flowers, and ImNet test sets respectively (clockwise, starting from top-left). No matter which dataset, the attention masks are semantically relevant, even when the correspondence is not one-to-one. More visualizations are given in the extended version [1].

weights. To encourage this behavior for the attention $\tilde{a}$, we also set $\Gamma = \Omega$, i.e., the key and query heads are the same. This way, in our trivial case, the attention is likely to be maximal for spatial locations that correspond, because $\boldsymbol{k}_{jm}^c$ and $\boldsymbol{q}_p$ will be the same for $p = m$.

For one experiment, we also augment the CrossTransformer with a global feature, which can help for some datasets like DTD (Describable Textures Dataset) with less spatial structure.

## 4 Experiments

We evaluate on Meta-Dataset [86], specifically the setting where the training is performed on the ImageNet train split only, which is 712 classes (plus 158 classes for validation, which are not used for training but only to perform early stopping). We then test on the remaining 130 held-out classes from ImageNet, as well as 9 other image datasets. Note that this is in contrast to another popular (and easier) setting, where the training also uses a subset of categories from more of these datasets: usually all datasets except Traffic Signs and COCO. For clarity, we'll use "Meta-Dataset Train-on-ILSVRC" to denote training on ImageNet only, and "Meta-Dataset Train-on-all" to denote when training occurs on more datasets. Test time consists of a series of episodes, each of which contains: (1) a support set between 50 and 500 labeled images which come from between 5 and 50 classes; and (2) an unlabeled query set with 10 images per class. Meta-Dataset aims for fine-grained recognition, so the classes in each episode are mutually similar: one episode may contain only musical instruments, another may contain only birds, etc.

Meta-Dataset is useful for studying transfer because different test datasets encapsulate different kinds of transfer challenges. For test datasets like CU-Birds [93], there are numerous similar classes in ImageNet train (20 bird classes in ImageNet train, versus 100 in the CU-birds test dataset). In contrast, for test datasets like Aircraft [53], there is just a single corresponding class in ImageNet train; therefore, algorithms which don't represent the intra-class variability for this class will be penalized. The ImageNet test set has images in a similar domain to the ImageNet train set but with different classes, while test datasets like COCO contain many similar classes to ImageNet, but with domain shift (in COCO, instances are generally not the subject of their photographs, and may be low-resolution or occluded). Finally, test datasets like OmniGlot combine these challenges, i.e., different classes in a substantially different domain.

### 4.1 Implementation details

To ensure comparability, we followed the public implementation of Prototypical Nets for Meta-Dataset [86] wherever possible. This includes using the same hyperparameters, unless otherwise noted. For the hyperparameters that were chosen with a sweep on the validation set (learning rate schedule and weight decay), we simply used the best values discovered for Prototypical Nets for all the experiments in this paper. See the extended version [1] for details of the CrossTransformer

architecture. We use no pretraining for CrossTransformers, although to be consistent with prior work [86] we use it for the experiments involving Prototypical Nets.

We incorporate two improvements from Meta-Baseline [17], which at test time is similar to Prototypical Nets (though it isn't trained as an episodic learner). The first is to keep exponential moving averages for Batch Norm statistics during training, and use those for Batch Norm at test time. Second, we note that Meta-Baseline does not train on fine-grained episodes sampled from the ImageNet hierarchy, as Prototypical Nets does, but rather on batches with uniformly-sampled classes. Empirically, Prototypical Nets trained only on fine-grained episodes struggle to do coarse-grained recognition, as required for datasets like COCO. Therefore, we only use the ImageNet hierarchy to make 50% of episodes fine-grained; the rest have categories sampled uniformly.

**Choice of network.** Prior implementations of networks like Prototypical Nets use relatively small networks (e.g., ResNet-18) with small input images (e.g. $126\times126$ pixels), and report that measures to increase capacity (e.g., Wide ResNets [100]) provide minimal benefits. This is surprising given that standard networks show improvements for increasing capacity (e.g., ResNet-34 outperforms ResNet-18 by 3% on ImageNet [38]). Making our networks spatially-aware requires higher-resolution, and also higher-capacity networks are especially important in self-supervised learning [24, 46]. Therefore, our experiments increase resolution to the standard $224\times224$ and use ResNet-34, and we also use normalized stochastic gradient descent [19, 59], which we found improved stability when fine-tuning more complex networks. Table 1 compares the Prototypical Nets performance of this network to that of using a ResNet-18. Increased capacity leads to only slight performance improvements, which are more pronounced for datasets that are similar to ImageNet; it harms, e.g., OmniGlot. Further details in the extended version [1].

For experiments with CrossTransformers, we also increased the resolution of the convolutional feature map by setting the stride of final block of the ResNet to 1, and using dilated convolutions to preserve the feature computation [33, 40]. This turns the usual $7\times7$ feature map for a $224\times224$ image into a $14\times14$ feature map. We ablate this choice in the extended version [1].

**Augmenting CTX with a global feature.** Recent works have also shown benefits for applying logistic regression (LR) at test time [83]. In practice, it is too expensive to apply LR to our query-aligned prototypes (as this would involve a separate classifier for every query). Therefore, we instead apply logistic regression to a globally-pooled feature and average the logits with those produced by the CrossTransformer. See the extended version [1] for details.

**Augmentation.** While most experiments use no augmentation (apart from SimCLR episodes) to be consistent with prior work [86], more recent work [17, 69, 83] showed that stronger data augmentation is effective. Therefore, for two experiments, we employ augmentation using the settings discovered in BOHB [69] (via Auto-Augment [20] on the validation set), with an extra stage that randomly downsamples and then upsamples images, which we find helpful as our network operates at higher resolution than many of the test datasets. This BOHB augmentation is only applied to the "MD-categorization" episodes, and not to the SimCLR episodes. Note this BOHB augmentation is different from SimCLR-style augmentation, which is used in SimCLR Episodes as well as in the ablation (SC-Aug) in Table 1. See the extended version [1] for details.

## 4.2 Results for self-supervised learning with SimCLR on Prototypical Nets

We first analyze the impact of SimCLR Episodes and other architectural choices in Table 1. For baseline Prototypical Nets, SimCLR Episodes generally improve performance, but this depends on architectural choices. Improvements are largest for datasets that are more distant from ImageNet, e.g., OmniGlot and Quickdraw, and datasets which require distinguishing between sub-categories ImageNet categories, e.g., Aircraft and Traffic Signs. In ImageNet, all commercial airplanes fall in a single ImageNet class; therefore, the success of SimCLR Episodes here suggests they recover features which are lost due to supervision collapse. Strangely, however, SimCLR Episodes interact with Batch Norm: we find more robust improvements when computing Batch Norm statistics from the test-time support set, but not when using exponential moving averages (EMA) as suggested by [17]. One possible interpretation is that the network has learned to use Batch Norm to communicate information across the batch: e.g., to distinguish between SimCLR Episodes and MD-categorization episodes. Using EMA at test time may prevent this, which may confuse the network. Interestingly,

Table 1: **Effects of architecture and SimCLR Episodes on Prototypical Nets, for Meta-Dataset Train-on-ILSVRC.** We ablate architectural choices: use of Exponential Moving Averages (EMA) at test time for Batch Norm (versus computing Batch Norm statistics on the support set at test time), image resolution (224, versus the baseline's 126), ResNet-34 (R34) replacing ResNet-18, SimCLR-style augmentation (SC-Aug), and the addition of 50% SimCLR Episodes (SC-Eps). The test datasets from Meta-Dataset are ImNet: Meta-Dataset's ImageNet Test classes; Omni: OmniGlot drawn characters; Acraft: Aircraft; Bird: CU-Birds; DTD: Textures; QDraw: Quick Draw drawings; Fungi: FGVCx fungi challenge; Flower: VGG Flowers; COCO: Microsoft COCO cropped objects. The best number in each column is bolded, along with others that are within a confidence interval [86]. Rank is the average rank for each method. Using SimCLR Episodes provides improvements on almost all datasets, and provides especially large boosts for datasets which are dissimilar from ImageNet, such as OmniGlot. However, simply using SimCLR transformations without instance discrimination (SC-Aug) harms results on almost all datasets. Increased capacity provides small benefits on some datasets, especially the more realistic and ImageNet-like datasets (e.g., birds), but actually harm others like OmniGlot. Note that in this table, QuickDraw uses the split from the original paper [86] rather than the (somewhat easier) split published for that paper's public benchmark. For all other tables, we use the split from the published benchmark.

| 224 | R34 | SC-Aug | SC-Eps | EMA | ImNet | Omni | Acraft | Bird | DTD | QDraw | Fungi | Flower | Sign | COCO | Rank |
|---|---|---|---|---|---|---|---|---|---|---|---|---|---|---|---|
| | | | | ✓ | 49.10 | 59.27 | 49.31 | 68.43 | 66.70 | 45.83 | 38.48 | 85.34 | 49.49 | 42.88 | 5.55 |
| | | | | | 49.77 | 55.70 | 52.06 | 68.58 | 67.27 | 49.86 | 37.68 | 84.32 | 50.27 | 41.92 | 5.20 |
| ✓ | ✓ | | | ✓ | 51.66 | 57.22 | 51.63 | 71.73 | **69.72** | 47.31 | **42.07** | 87.29 | 47.45 | **44.38** | 4.35 |
| ✓ | ✓ | | | | **52.51** | 49.87 | 56.47 | **72.81** | 68.45 | 51.41 | **42.16** | **87.92** | 54.40 | 40.60 | 3.30 |
| ✓ | ✓ | ✓ | | ✓ | 47.58 | 55.73 | 46.93 | 57.75 | 54.88 | 42.91 | 37.42 | 83.82 | 46.88 | **43.36** | 7.55 |
| ✓ | ✓ | ✓ | | | 47.94 | 51.79 | 54.58 | 62.84 | 58.64 | 46.36 | 36.06 | 76.88 | 48.35 | 38.77 | 7.45 |
| ✓ | ✓ | | ✓ | ✓ | 49.67 | 65.21 | 54.46 | 60.94 | 63.96 | 50.64 | 37.84 | **88.70** | 51.61 | **42.97** | 4.35 |
| ✓ | ✓ | | ✓ | | **53.69** | **68.50** | **58.04** | **74.07** | **68.76** | **53.30** | 40.73 | 86.96 | **58.11** | 41.70 | **1.90** |
| ProtoNets [86] | | | | | 50.50 | 59.98 | 53.10 | 68.79 | 66.56 | 48.96 | 39.71 | 85.27 | 47.12 | 41.00 | 5.35 |

we will show later that SimCLR Episodes don't harm CrossTransformers as they harm Prototypical Nets when using EMA at test time, suggesting the two architectures solve the problem differently.

Recall that converting an MD-categorization episode into a SimCLR episode makes two changes to the episode: it 1) applies data augmentation, and 2) converts the classification problem to "instance discrimination," by selecting images from the support set as a new query set, and requiring the network to predict the selected indices. To ensure that we are not simply seeing the effect of data augmentation, we also implemented a baseline (SC-Aug) that does 1 but not 2 to the input MD-categorization episodes, and does this augmentation for all episodes (rather than 50%, which is the fraction of MD-categorization episodes that are converted to SimCLR episodes for SC-Eps experiments). Indeed, we see no improvements for this change, and in fact non-trivial performance loss from this, mirroring the result for supervised learning in the original paper [16]. This reinforces that SimCLR was designed for self-supervised learning, and so the transformations are more severe than is usually optimal for supervised learning.

Finally, we see small improvements from using larger networks and higher resolution for the baseline model. While our baseline is overall better than the baseline Prototypical Nets implementation [86], it is still below the state-of-the-art for centroid-based methods which rely more heavily on pretraining, and use no episodic training [17].

### 4.3 CrossTransformers results

Given these performant features, we next turn to CrossTransformers. Table 2 compares CrossTransformers (CTX) with and without SimCLR episodes to several state-of-the-art methods, including the Prototypical Nets on which they are based. We see that CrossTransformers provide strong performance on their own, including having a better average rank than all baselines. With SimCLR episodes providing more versatile features, we see further improvements, with performance on-par or better than the best methods on almost every dataset. We note particularly large improvements on OmniGlot, which has a large domain gap relative to the training data. We also see strong improvements on Street Signs, Aircraft, and Flowers, where multiple test-time categories map to few training-time categories, and often exhibit well-defined spatial correspondence.

Table 2: **CrossTransformers (CTX) comparison to state-of-the-art.** We compare four versions of CrossTransformers to several state-of-the-art methods, which are the best performers among those evaluated for Meta-Dataset Train-on-ILSVRC. We see that CTX alone has better average rank than any baseline. Adding SimCLR episodes (+SimCLR Eps) and data augmentation inspired by BOHB [69] (+Aug) further improves results. Our full model is on-par or above prior methods on all but one dataset, sometimes with large gaps over the best baseline (e.g., +5% on OmniGlot, +13% on Aircraft, +5% on Signs), and furthermore, each prior method has some datasets where we outperform by a larger margin (the next best average rank [83], performs 19% worse on Aircraft and 17% worse on OmniGlot). Finally, adding a test-time Logistic Regression classifier inspired by Tian et al. [83] improves performance on the one dataset—DTD textures—that was otherwise lacking. Note that most of these methods [17, 69, 83] are unpublished concurrent work.

| | ImNet | Omni | Acraft | Bird | DTD | QDraw | Fungi | Flower | Sign | COCO | Rank |
|---|---|---|---|---|---|---|---|---|---|---|---|
| Finetuning [86] | 45.78 | 60.85 | 68.69 | 57.31 | 69.05 | 42.60 | 38.20 | 85.51 | 66.79 | 34.86 | 12.20 |
| ProtoNets [86] | 50.50 | 59.98 | 53.10 | 68.79 | 66.56 | 48.96 | 39.71 | 85.27 | 47.12 | 41.00 | 12.65 |
| ProtoNets+MAML [86] | 49.53 | 63.37 | 55.95 | 68.66 | 66.49 | 51.52 | 39.96 | 87.15 | 48.83 | 43.74 | 11.55 |
| CNAPS [66] | 50.60 | 45.20 | 36.00 | 60.70 | 67.50 | 42.30 | 30.10 | 70.70 | 53.30 | 45.20 | 13.55 |
| BOHB-L [69] | 50.60 | 64.09 | 57.36 | 67.68 | 70.38 | 46.26 | 33.82 | 85.51 | 55.17 | 41.58 | 11.50 |
| BOHB-NC [69] | 51.92 | 67.57 | 54.12 | 70.69 | 68.34 | 50.33 | 41.38 | 87.34 | 51.80 | 48.03 | 10.15 |
| BOHB-NC Ensemble [69] | 55.39 | 77.45 | 60.85 | 73.56 | 72.86 | 61.16 | 44.54 | 90.62 | 57.53 | 51.86 | 7.45 |
| Dhillon et al. [21] | - | - | 68.69 | 74.26 | 77.35 | - | - | 88.14 | 55.98 | 40.62 | - |
| Meta-Baseline [17] | 59.20 | 69.10 | 54.10 | 77.30 | 76.00 | 57.30 | 45.40 | 89.60 | 66.20 | 55.70 | 7.20 |
| Tian et al. LR [83] | 60.14 | 64.92 | 63.12 | 77.69 | 78.59 | 62.48 | 47.12 | 91.60 | 77.51 | 57.00 | 5.50 |
| Tian et al. LR-distill [83] | 61.58 | 64.31 | 62.32 | 79.47 | **79.28** | 60.83 | 48.53 | 91.00 | 76.33 | **59.28** | 4.60 |
| ProtoNets (Our implementation) | 51.66 | 57.22 | 51.63 | 71.73 | 69.72 | 53.81 | 42.07 | 87.29 | 47.45 | 44.38 | 11.10 |
| CTX | 61.94 | 76.52 | 79.65 | **84.06** | 76.26 | 65.67 | **52.53** | 94.11 | 70.47 | 53.51 | 3.85 |
| CTX+SimCLR Eps | **63.79** | 80.83 | **82.05** | 82.01 | 75.76 | 68.84 | 52.01 | 94.62 | 75.01 | 52.76 | 3.05 |
| CTX+SimCLR Eps+Aug | **62.76** | 82.21 | 79.49 | 80.63 | 75.57 | **72.68** | 51.58 | 95.34 | 82.65 | 59.90 | **2.25** |
| CTX+SimCLR Eps+Aug+LR | 62.25 | **82.03** | 77.41 | 76.66 | **80.29** | 72.24 | 49.39 | 93.05 | 75.25 | **60.35** | 3.40 |

DTD, however, is more challenging for basic CTX, which is unsurprising since textures have little of the kind of spatial correspondence that CTX attempts to find. COCO is also challenging, likely due to its extremely large intra-class variation (e.g., occlusion) and the fact that many categories overlap with ImageNet-train categories, meaning that simply memorizing categories from the training set may be more useful than using test-time appearance. To explore this trade-off, we applied logistic regression at test time to a globally pooled feature (see the extended version [1]), which provides additional logits that are averaged with the CTX logits. We see non-trivial improvements on DTD by using this, although we sacrifice some performance on other datasets, such as Signs and Aircraft. This implies that there's a fundamental tension between learning categories based on global features, and decomposing the task into local features. More work is needed to better combine these two ideas.

Finally, Figure 3 depicts the correspondence inferred by the CrossTransformer. The attention is often semantically meaningful: object parts are well matched, including heads, bodies, feet, engines, and strings. The attention is often not one-to-one either: for the flower, the single query flower is matched to multiple flowers in some of the support images. Furthermore, the matching works even when the fine-grained classes are not the same, such as the different species of birds, suggesting that the attention is indeed a coarse-grained matching that has not overfit to the training-set classes.

## 5   Conclusion

Within a single domain, deep networks have a remarkable ability to compose and reuse features in order to achieve statistical efficiency. However, this work shows the hidden problem with such systems: the networks compose features in a way that conflates images which have different appearance but the same label, i.e., it loses information about intra-class variation that may be necessary to understand novel classes. We propose two techniques that help resolve this problem: self-supervised learning, which prevents features from losing that intra-class variation, and CrossTransformers, which help neural networks classify images using local features that are more likely to generalize. However, this problem is far from resolved. In particular, our algorithm provides less benefit when less spatial structure is available, when knowledge of train-time categories can be useful (as in, e.g., COCO), or when higher-level reasoning is required (e.g., finding conjunctions of multiple objects). Allowing this algorithm to use spatial structure only where relevant remains an open problem.

## Broader Impact

The algorithm presented in this paper most directly applies to few-shot recognition, which has numerous uses in industry, including vision systems for robotics that must adapt to new objects, and photo-organizing software which must infer the presence of new classes of objects on-the-fly. Unfamiliar objects are ubiquitous in many real-world vision applications due to the so-called 'long tail' [95] of objects that occur in real scenes, and therefore we expect our algorithm to improve the robustness of visual recognition systems. While our current work only addresses classification, many other tasks in computer vision, such as object detection and segmentation, use neural network representations that can likewise be made more robust using the kind of architectures presented here.

Our algorithm attempts to build representations which factorize the object recognition problem into sub-problems (feature correspondence and feature comparison) that will each transfer correctly to new datasets. We hope that further research in this direction may help address dataset biases, including biases regarding race, gender, or other attributes [98], by helping to disentangle the truly meaningful traits from the spurious correlations. Finally, while this algorithm presents an advance to state-of-the-art in understanding rare objects, the general performance of such systems is still far below human performance. For safety-critical applications (e.g., surgery or self-driving cars), relying on the ability of vision systems to correctly interpret unusual situations is risky with current systems, even with the advances presented here.

## Funding Disclosure

This work was funded by DeepMind.

## Acknowledgments

The authors would like to thank Pascal Lamblin for help with Meta-Dataset, Olivier Hénaff for help with SimCLR, Yonglong Tian for help in reproducing baselines, and Relja Arandjelović for invaluable advice on the paper. They are also grateful to Jean-Baptiste Alayrac, Joao Carreira, Mateusz Malinowski, Viorica Pătrăucean, Adria Recasens, and Lucas Smaira for helpful discussions, support, and feedback on the project.

## Footnotes

[1]We enforce that the sampled queries have the same class distribution as $Q$, and have no repeats.

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
