[Reviews · NeurIPS 2020]

Review 1

Summary and Contributions: Few-shot learning is a challenging problem, which requires performing a supervised learning task with a small labeled support set. The classic few-shot learning problem has been extended (so to say) for an episodic learning setting by Triantafillou et al. The proposed cross-transformer framework addresses this episodic few-shot learning problem. The approach builds on the Prototypical Nets (Snell et al). Specifically, it makes the following contributions Identification of supervision collapse. It is empirically observed that the Prototypical Nets do not learn about representations outside the set of training classes. The authors propose an episodic version of SimCLR (Chen et al) to learn embeddings that discriminate between every image in the dataset while being invariant to image transformations. This involves learning a “pretext” task, which is a |S|- way classification for each query, where the query set consists of images from the support set having undergone a random transformation. This pretext task is along the lines of the work of Chen et al. The authors claim that the episodic version of SimCLR helps to overcome the problem of supervision collapse. The second contribution is a novel architecture called CrossTransformers based on the work on attention by Vaswani et al. The proposed framework allows for ‘local-part’ based comparison of representations in the support and query images. The corresponding local-parts are estimated through an attention mechanism. This allows for a more fine-grained comparison of image features that are (as put forth by the authors) agnostic to the classes. The central idea is that different classes can have similar ‘local-parts’ and these features can be learned through the proposed framework. The proposed contributions have been tested on the recently proposed Meta-Dataset. Investigations have been performed to understand and quantify the impact of both the episodic SimCLR learning and the Cross Transformer framework. [post rebuttal] I thank the authors for the clarification. I believe that my concerns have been addressed largely and I am satisfied with the response. I retain my original rating.

Strengths: The first contribution is a direct application of SimCLR approach. However, I find the discussion on supervised collapse quite interesting. The authors have identified a (non-trivial) problem in the current approach, and have demonstrated the effectiveness of an existing approach SimCLR with some modifications. The modifications are perhaps a little more straight forward. Even though as I mention in the weakness section, there are elements that need more investigations, nevertheless, this does not diminish the importance of the first contribution. The second contribution is a non-trivial extension of the Transformers framework of Vaswani et al. The significance of the contribution lies in its ability to learn local patterns across images of different classes that aid in the generalization of the model. I have not seen similar work in the literature on few-shot learning. The quantitative and qualitative results support the effectiveness of the approach.

Weaknesses: As I understand the supervision collapse refers to the inability of a model to learn meaningful representations for samples beyond the samples of classes belonging to the training set. How does the proposed self-supervision training help in overcoming this problem? The SimCLR approach can learn to discriminate images between a single class. This is enforced in the SimCLR episodes. However, how does this help to learn information about other classes? Isn’t this more important for the few-shot learning setting? The discussion in Section 3 of the supplementary material (especially the nearest neighbor experiment) can be improved. While the problem of supervision collapse is aptly demonstrated, can you please show some evidence that the SimCLR episodic learning helps to overcome it? Specifically, redo the nearest neighbor experiments for the representation learned through SimCLR episodes and investigate changes in the distribution? The description of the attention mechanism, specifically the Key Head, in the Cross-Transformer framework is envisioned to be different for both support and query set. However, towards the end (line 176) it is mentioned that the two heads are the same. The rationale behind having the same Key and Value head across the support and query sets is unclear. In particular, how does this choice help the d function to behave like a distance metric? The attention mechanism is quite complex. A discussion on the complexity of the model, especially in terms of computations and the improvement in the performance will help the readers better understand the strengths of the approach. Typical few-shot learning settings deal with very few training samples per class (less than 10). However, the experiments conducted in this paper deal with 5-50 instances per class. If there are close to 50 instances, then I wonder if it still can be considered as the few-shot setting. I do believe there is no clear-cut answer to what is ‘few’ in few-shot learning. But my point is to encourage the authors to conduct experiments with varying numbers of ‘few’ samples. Minor comments Lines 60-65 are redundant. They can be replaced with some other material. Line 142 - in the mathematical expression, the distance should be computed between a transformed instance from the support set and the query set. Perhaps one of the x_q in the numerator should be replaced with x_i?

Correctness: The methodology appears to be technically correct.

Clarity: The paper is well-written and easy to follow.

Relation to Prior Work: The authors have cited most of the relevant work and have also clearly differentiated the proposed approach with those in the literature.

Reproducibility: Yes

Additional Feedback:


Review 2

Summary and Contributions: The authors propose two methods to mitigate the supervision collapse problem in a few-shot image classification, where any information unnecessary for the (pre-)training task is lost. First, they propose to use SimCLR based self-supervised learning to encourage learning more generalized features. Second, they propose a novel CrossTransformer model architecture which can find coarse spatial correspondence between the query and labeled images. They demonstrate the effectiveness of proposed methods through Meta-Dataset evaluations and showed they can outperform previous state-of-the-art methods by a significant margin.

Strengths: (1) Novelty and Significance To the best of my knowledge, the proposed CrossTransformer is a novel model architecture that can learn soft correspondences of high-level spatial features in query and support set images (Disclaimer: I am not a computer vision expert, so I will rely on other reviewers). The updated results in the supplementary material are impressive and the significant margin compared to previous state-of-the-arts would benefit the broad NeurIPS community. (2) Motivation and Implementation details The paper stems from good motivation (supervision collapse), and the authors provide great effort to show it is an important problem. Although codes are not provided, I could also find the authors’ effort to provide detailed implementation and experiment information for the reproducibility of results.

Weaknesses: (1) Differences compared to the original SimCLR algorithm Although the authors claim that they modified the SimCLR algorithm (page2 line60), I could not quite understand the main differences. At the end of the section3.1, they stated if we use Cosine Distance, the loss would be identical to SimLCR. In Table1, I was confused that “SimCLR Baseline” might be the results of the original SimCLR (rather than the modified one), but it seems like it is just the results from SimCLR’s random transformations. Please correct me if I misunderstood, or change the naming to prevent confusion. In summary, what is the main contribution regarding SimCLR? Is it a straight-forward application or are there important changes? (2) Compared baselines Although the proposed method is particularly relevant to pre-trained image representations in self-supervised fashion, none of them are included in the comparative analysis. Since the goal of image representation pre-training is to learn generalizable features, I think they would also be able to provide strong baselines for a few-shot learning setting. Additionally, I think the following paper should also be referenced and compared in the manuscript (“A baseline for few-shot image classification”, ICLR 2020). It was presented in the same conference as the Meta-Dataset paper.

Correctness: The claims and methods seem correct.

Clarity: The paper is generally well written and easy to understand. The contribution is clear, and the way the authors claim is well understandable and notations are explained well.

Relation to Prior Work: As I mentioned in my previous comments, it requires more clear explanation and comparative experiments regarding other self-supervised pre-training methods.

Reproducibility: Yes

Additional Feedback: No additional comments. ---- Post Author Feedback Comments --- I have read the Author Feedback and I'm happy that the authors have clarified some points.


Review 3

Summary and Contributions: This paper proposed two methods to increase the adaptability of neural network representations and used few-shot image classification as testbed to validate their effectiveness. The first is to leverage the self-supervised learning task "instance discrimination" (or contrastive learning); an additional cross-transformer is proposed for local correspondence between query image and support images. ========== update after rebuttal ================== Firstly, I appreciate authors' efforts on addressing my concerns. I would like to upgrade my score a bit. But concerns still holds. I still feel the comparisons here are apple to orange, for example in the table in the rebuttal, I think your method are using advanced data augmentation techniques, such as Auto-augment, but how about other methods? It seems the baseline of 'chen et. al.' only used simple augmentation. Besides, how to improve from the results in the original submission to the new number remains mysterious to me (which actually is quite important).

Strengths: The cross-transformer part is interesting. Though it is motivated by the well-known transformer model, this is the first application on few-shot learning. Therefore it does have some novelty for this part. The evaluation is conducted on Meta-dataset, and it seems to achieve SoTA results, if we don't count in arXiv papers that are 3-4 months earlier than the NeurIPS deadline. (Personally I think they should be counted, see below)

Weaknesses: The comparison with previous work is apple-to-orange: - Most of previous methods from Meta-Dataset use ResNet-18 operating on resolution 128, while CTX here uses ResNet-34 and works with resolution of 224. Definitely more computational footprint. - While you are showing that this (ResNet-34 + 224 + N.SGD) does not improve ProtoNet, it doesn't indicate it can not improve others. Especially the ProtoNet present in your paper is a bit suspicious, i.e., in 230-231, you claim that data-augmentation harms, which is counter-intuitive and you don't perform enough analysis but just blame that the augmentation is self-supervised specific. To me, this is not really true, because SimCLR aug only adds Gaussian Blur and Color jittering on top of very standard augmentation. - Using Auto-augment is problematic. The policy in Auto-augment is developed by seeing the labels of all 1k classes within ImageNet. While it's fine to say that you only use 712 classes for training, such claim is not that "clean". Important details are missing in the main text, for example, the authors never mention that they use the standard classification training in the main text. However recent studies [a,b] found this part is actually crucial (and the authors should definitely discuss these two papers as they are very relevant). They authors neither discuss the effects of pre-training, nor mention you use two-stage training in the main text (only exists in supp.) [a] A New Meta-Baseline for Few-Shot Learning [b] Rethinking Few-Shot Image Classification: a Good Embedding Is All You Need? The results present in [a] and [b], which only use pre-trained features, are much better than CTX-SimCLR eps +Aug here, which uses both pre-training and episode learning. Not to mention that this paper uses a larger network as well as 224 resolution. I also wonder if the improvement comes from longer training schedule because of additional SimCLR episode? There is no such ablation. You should compare: - ProtoNet with X episodes - Your method with 0.5*X normal episode + 0.5*X SimCLR episode.

Correctness: Technically this paper looks good to me.

Clarity: Well-written and easy to follow. A tiny inconsistency: line 164 uses x_q, while line 170 uses x^q

Relation to Prior Work: I am not satisfied with the related work section, though this paper does cite many previous works. Firstly, the authors cite too many less relevant works. For example: - Too many correspondence work are included, can you focus on discussing the most relevant ones in depth? - While the authors argue that they use "self-supervised learning" to avoid supervision collapse, I wonder if it's the specific task of contrastive learning that is helping, or other self-supervised tasks such as rotation, jigsaw, and colorization also helps? As far as I see, it's more about the instance discrimination task or contrastive learning. To this end, the following work are more relevant than rotation, jigsaw, and colorization (and I think they are more relevant and it's good to discuss these works in terms of contrastive learning): - "Unsupervised feature learning via non-parametric instance discrimination" - "Contrastive multiview coding" - "Self-supervised learning of pretext-invariant representations" Also the following paper that explicitly discussing self-supervised learning for few-shot learning is missing: - When Does Self-supervision Improve Few-shot Learning? Missing relevant few-shot learning papers: - Low-shot learning from imaginary data - Meta-learning to detect rare objects - Learning compositional representations for few-shot recognition

Reproducibility: No

Additional Feedback: See above


Review 4

Summary and Contributions: The paper aims at using a attention-based model to solve the few-shot image classification problem on a recently released dataset that explores different settings. They show how their model is able to exploit spatial context using the attention mechanism and to incorporate this information in the distance calculation between classes.

Strengths: The paper makes use of attention mechanism to build a context-aware distance between the query image, which has to be classified, and the possible image sets that define different classes, some of which are unseen during training. While the attention mechanism is not a novelty, the way it has been used for this task is interesting and original enough to be considered as a contribution in the few-shot learning field. The solution proposed is elegant in its simplicity and is based on a well-defined mechanism which has showed its usefulness and soundness in past works. The experiments are built on top of the recently released meta-dataset, which define a standard protocol to evaluate performances on few-shot learning task in many different setting. The empirical results show the validity of the approach.

Weaknesses: The method distance is based on whether the images in the support set have matching characteristic to the query image and that’s a good approach; the problem is that this matching is only one directional from query to support set and not viceversa. This fails will when there are wrong classes that also contains those feature in addition to other features that should help distinguish them. Let’s assume for example to have a support set for the class of “green triangle” and a support set for “green triangle and yellow triangle”. A query for a green triangle image will match to both set and not only to the green triangle class even if the “green triangle and yellow triangle” support set is also defined by the characteristic of the yellow triangle. There are no qualitative results in the main paper only quantitative. No study on failed cases and why the attention mechanism did fail to give a correct class

Correctness: The method is clear, simple, built on top of a well defined mechanism and proven to be effective in solving the task

Clarity: The paper is well written and manages to express clearly the scope and results

Relation to Prior Work: The work improve on past works by introducing the concept of using the attention mechanism to spatially match local features in the query image to those in the support set and build a distance metric that takes this information into account.

Reproducibility: Yes

Additional Feedback: The name of the approach needs to be changed; the Transformer model is a well-defined layered encoder-decoder model which is built on top of the self-attention mechanism that produces sequences as output. The method proposed does not have layers, does not have an encoder-decoder structure, does not produce a sequence as output; the only thing that has in common with the Transformer model is the use of attention mechanism. Calling this approach CrossTransformer when it does not have any of the characteristics with the Transformer model is very misleading. UPDATE: The rebuttal satisfied me.

[Author Response · NeurIPS 2020]

We thank our reviewers for their insightful comments, and address their remarks below.
**R1: Why SimCLR Episodes help supervision collapse?** For example, in the ImageNet training set, all airplanes are
a single category. Therefore, for ProtoNets, the loss is minimized if all aircraft have identical features, which makes it
difficult to tell different models apart when evaluating on Aircraft. SimCLR Episodes prevent this collapse. SimCLR
Episodes do improve nearest neighbor results: proportion of test images with 1 or more correct matches improves from
34.1% to 48.8%, and for those with 2 or more neighbors from the same train set class decreases from 55.3% to 43.3%.
**R1: Shared heads.** Distances between similar images should be small. Shared heads achieve this as they yield similar
representations (both keys and values) for similar support and query images, even with little training.
**R1: Few-shot.** We followed the standard procedure for Meta-Dataset, as we were particularly interested in the kind of
transfer challenges it poses. However, we agree that k-shot is also interesting and will include it.
**R2: Differences with SimCLR.** The original SimCLR is proposed as a pre-training method, while SimCLR Episodes
reformulate SimCLR to work in an *episodic training setting* with instance discrimination over the support set images.
We will improve the "SimCLR Baseline" name.
**R2: Baselines.** We tried self-supervised pre-training (rotations and SimCLR) and found like other works [3] that it
doesn't improve over supervised pre-training. [2] has comparable numbers only on 6 datasets: 2 datasets are missing
and 2 ignore the fine-grained split; for the remainder, we have better results on all but DTD. We will include these results.

| | ImNet | Omni | Acraft | Bird | DTD | QDraw | Fungi | Flower | Sign | COCO |
|---|---|---|---|---|---|---|---|---|---|---|
| ProtoNet+R34+224+N.SGD **2x** | 50.50 | 51.33 | 51.22 | 73.89 | 66.16 | 46.04 | 40.58 | 84.76 | 49.54 | 39.98 |
| Chen et al. [1] orig | 59.20 | 69.10 | 54.10 | 77.30 | 76.00 | 57.30 | 45.40 | 89.60 | 66.20 | 55.70 |
| Chen et al. [1]+R34+224 | 64.41 | 61.22 | 56.09 | 80.69 | 78.86 | 54.89 | 47.29 | 90.00 | 68.90 | 58.60 |
| Tian et al. LR [4] | 60.14 | 64.92 | 63.12 | 77.69 | 78.59 | 62.48 | 47.12 | 91.60 | 77.51 | 57.00 |
| Tian et al. LR-distill [4] | 61.58 | 64.31 | 62.32 | 79.47 | 79.28 | 60.83 | 48.53 | 91.00 | 76.33 | 59.28 |
| CTX | 61.53 | 73.34 | 72.32 | 80.83 | 72.25 | 55.61 | 49.54 | 93.16 | 66.02 | 51.22 |
| CTX+SimCLR Eps | 61.54 | 81.88 | 81.53 | 80.30 | 75.64 | 59.91 | 49.76 | 94.19 | 77.00 | 53.48 |
| CTX+SimCLR Eps+Aug | 61.42 | 82.75 | 81.84 | 77.98 | 74.77 | 64.80 | 49.95 | 95.28 | 84.80 | 57.87 |

**R3: Baselines.** We agree that [1] and [4] are interesting (although they added Meta-Dataset results on April 1 and
June 17, respectively, not "3-4 months earlier than the [June 5] NeurIPS deadline"). They engineered two key aspects of
the training which we did not explore: (1) During training, categories are sampled *uniformly* from ImageNet (rather
than sampling episodes which contain related categories); we incorporate this by sampling 50% of episodes where
the category distribution is also sampled uniformly from ImageNet. (2) They use training moving averaged batch
norm statistics for test episodes, rather than using batch norm statistics from the test-time support set. The table shows
updated results incorporating these ideas; we outperform or match (within the confidence interval) the cited works on all
datasets but DTD (which has no spatial structure for CTX to exploit). Note the especially large gap for Omni & Acraft.
**R3: Apples & oranges.** We are also surprised that larger networks and higher resolution make little difference. We
reproduced the result by running the open-source code for the latest SotA method of Chen et al. [1] with ResNet-34
and $224 \times 224$ images and found only small improvements (1-3%) over the original ResNet18-128px on most datasets
(see table), while harming datasets with a larger domain gap (Omniglot & QuickDraw). The "suspicious" loss of
performance from data augmentation is also reported in Table 1 of the original SimCLR paper.
**R3: Auto-Augment.** CTX+Aug experiments used the exact augmentation settings described in prior work (BOHB).
BOHB used only the Meta-Dataset's ILSVRC validation classes for tuning the augmentation, *not* all 1k classes.
**R3: Two-stage training.** We follow Meta-Datasets' standard training procedure with classification-based pre-training.
**R3: Reproducibility.** We will release the full code to enable reproducibility.
**R3: Longer training schedule.** As suggested, we re-ran ProtoNets doubling the number of episodes and doubling the
episodes between LR decays ("2x" in table above). We saw no improvement.
**R3: Related work.** Thanks for the suggestions. Work on self-supervised (SSL) for few-shot was omitted due to an
editing error. This prior work uses auxiliary losses and networks, whereas ours directly integrates SSL into episodic
training. We will improve our discussion on SSL, including contrastive methods, and add the suggested few-shot papers.
**R4: Failure cases.** We agree that the example of green and red triangles would be a challenge for our algorithm.
However, such cases are rare in Meta-Datataset, since the dataset is mostly about distinguishing between fine-grained
object categories. Still, we agree this point is worth discussing. We will include this along with more qualitative
examples of failures.
**R4: Naming.** We agree that the transformer naming convention is less-than-ideal, but computer vision now has a
tradition of using 'transformer' to refer to only the attention mechanism: e.g., Fang et al. "Scene memory transformer
for embodied agents in long-horizon tasks," Girdhar et al. "Video action transformer network." Using this name will
help others working on similar models find our work.

[1] Y. Chen et al. A new meta-baseline for few-shot learning. *arXiv preprint arXiv:2003.04390*, 2020.
[2] G. S. Dhillon et al. A baseline for few-shot image classification. *Proc. ICLR*, 2020.
[3] J.-C. Su, S. Maji, and B. Hariharan. When does self-supervision improve few-shot learning? In *Proc. ECCV*, 2020.
[4] Y. Tian et al. Rethinking few-shot image classification: a good embedding is all you need? In *Proc. ECCV*, 2020.


[Meta-Review · NeurIPS 2020]

The reviewers appreciate the discussion and insight given on the supervision collapse problem and the new cross-transformer technique. While there have been some concerns that recently published simpler approaches have better performance, the authors provided improved results in the rebuttal that still show the advantage of the proposed technique.